

# UV-B irradiation promotes anthocyanin biosynthesis in the leaves of *Lycium ruthenicum Murray*

Shengrong Chen[1], Yunzhang Xu[1], Weimin Zhao[2], Guomin Shi[3], Shuai Wang[1] and Tao He[1]

[1] State Key Laboratory of Plateau Ecology and Agriculture, Qinghai University, Xining, Qinghai, China
[2] School of Ecol-Environmental Engineering, Qinghai University, Xining, Qinghai, China
[3] College of Agriculture and Animal Husbandry, Qinghai University, Xining, Qinghai, China

## ABSTRACT

Anthocyanins are the most valuable pigments in *Lycium ruthenicum Murray* (*L. ruthenicum*). Although ultraviolet-B (UV-B) irradiation is a key environmental factor influencing anthocyanin biosynthesis in *L. ruthenicum*, the deep molecular mechanism remains unclear. Herein, we examined the changes in the total anthocyanin content and transcriptomic characteristics of *L. ruthenicum* leaves following UV-B irradiation treatment. The results showed a twofold increase in anthocyanin content in the leaves of *L. ruthenicum* after the treatment. The transcriptome analysis showed that the expression of 24 structural genes identified in the anthocyanin synthesis pathway was up-regulated. In particular, F3′H (Unigene0009145) and C4H (Unigene0046607) exhibit notable up-regulation, suggesting their potential roles in anthocyanin synthesis. Protein interaction network results revealed that MYB1 (Unigene0047706) had the highest connectivity, followed by bHLH (Unigene0014085). Additionally, UVR8 (Unigene0067978) and COP1 (Unigene0008780) were found to be highly involved in UV-B signal transduction. These findings provide new insights into the genetic and biochemical mechanisms that regulate anthocyanin production, and could guide agricultural practices to reduce environmental impacts and improve crop yield and quality.

## INTRODUCTION

Anthocyanins are pigments responsible for red, purple, and blue colors in plants, crucial for attracting pollinators and seed dispersers, aiding in plant reproduction. They protect plants against environmental stresses like UV radiation, cold, and oxidative stress, thanks to their ability to absorb UV light and their antioxidant properties (*Neill & Gould, 2003*; *Landi, Tattini & Gould, 2015*). Anthocyanins also facilitate plant signaling and communication, deterring herbivores and influencing interactions with other plants and microorganisms. These multifaceted roles make anthocyanins essential for plant survival, reproduction, and ecological interactions, contributing significantly to their overall health and vitality (*Liu et al., 2018*; *Cappellini et al., 2021*; *Sharma et al., 2022*). *L. ruthenicum* is an important

Corresponding author
Tao He, hetaoxn@aliyun.com

ecological and economic species in the Qinghai-Tibet Plateau. The leaves of *L. ruthenicum* are a byproduct of the planting industry and contain functional components similar to its fruits. Therefore, the leaves are commonly used for both medicinal and culinary purposes due to their nutritional value and cost-effectiveness (*Sharma et al., 2022*) . The utilization of the leaves has also increased the value of the *L. ruthenicum*. Anthocyanins are valuable functional components found in the leaves of *L. ruthenicum*. These active substances have various physiological benefits, such as antioxidant, anti-cancer, and anti-diabetic properties (*Zheng et al., 2011*; *Liu et al., 2020*; *Cappellini et al., 2021*).

UV-B irradiation has various physiological and biochemical effects on plants, impacting their growth, development, and metabolic processes (*Jenkins, 2009*). Plants detect UV-B radiation through photoreceptors and transmit signals to the nucleus, leading to the expression of genes related to anthocyanin synthesis. This, in turn, regulates anthocyanin synthesis in plants (*Takshak & Agrawal, 2019*; *Li et al., 2023*). Specifically, UV-B irradiation can induce the expression of transcription factors for anthocyanin synthesis, such as SIBBX20/SIBBX21, MYB75/PAP1, HY5, COP1, and WRKY11 (*Liu et al., 2019*). These genes directly or indirectly regulate downstream structural genes, promoting transcription of genes such as PAL, C4H, 4CL, CHS, CHI, F3H, FLS, DFR, ANS, and UFGT. These genes encode enzymes involved in the anthocyanin synthesis pathway. The increased expression of the enzymes leads to a subsequent increase in anthocyanin content (*Chaves-Silva et al., 2018*; *Liu et al., 2018*).

Research has shown that MdBBX20 interacts with MdHY5 under UV-B irradiation and can bind to the promoters of the biosynthesis genes MdDFR and MdANS, regulating their expression and promoting anthocyanin synthesis in *apples* (*Fang et al., 2019*). Additionally, UV-B induces the expression of MdWRKY71-L, which directly regulates anthocyanin biosynthesis by forming a transcriptional complex with MdHY5-MdMYB1 and interacting with the MdUFGT promoter (*Su et al., 2022*). In addition, research on *Arabidopsis thaliana* has shown that MYB13 plays a positive role in UV-B-induced cotyledon expansion in a UVR8-dependent manner. MYB13 binds directly to the promoters of CHS, CHI, and FLS, positively regulating their expression for flavonoid accumulation and UV-B tolerance (*Qian et al., 2020*). Similarly, studies have found that under long-term UV-B irradiation, the expression of transcription factors such as UVR8, COP1, HY5, and MYB, as well as structural genes like FLS and F3'H, increased in *Ginkgo biloba* leaves, leading to an increase in flavonol content (*Zhao et al., 2020*). In particular COP1 and HY5 are key regulators in UV-B signaling. COP1 is an E3 ubiquitin ligase that mediates the degradation of photomorphogenesis-promoting proteins in darkness, while HY5 is a bZIP transcription factor essential for light-responsive gene expression. Under UV-B irradiation, COP1's activity is inhibited, leading to the stabilization of HY5, which then promotes the expression of UV-B-responsive genes. This interaction is crucial for understanding how plants adapt to UV-B stress, guiding our study on anthocyanin synthesis in the leaves of *L. ruthenicum*.

*L. ruthenicum* also exhibits a significant response to UV-B irradiation. Transcriptome analysis revealed a significant increase in the expression of the UDP-glucose flavonoid 3-O-glucosyltransferase (UFGalT) gene, which is related to anthocyanin synthesis, in the leaves of *L. ruthenicum* after UV-B irradiation (*Chen et al., 2015*). However, there

has been a lack of systematic research on genes related to anthocyanin synthesis in the leaves of *L. ruthenicum* under UV-B irradiation. The molecular mechanism by which UV-B promotes anthocyanin accumulation has not been well understood. The research results can reveal new protection strategies and regulatory pathways, thereby optimizing cultivation methods and improving the yield and quality of anthocyanins.

In this paper, we studied the effects of UV-B irradiation treatment on the anthocyanin biosynthesis in *L. ruthenicum* leaves using transcriptome analysis. The study investigated the changes in anthocyanin content caused by UV-B irradiation and identified key regulatory genes that promote anthocyanin accumulation in the leaves. The findings provide fundamental data for future molecular biology research on promoting anthocyanin accumulation in *L. ruthenicum* under UV-B irradiation. Additionally, the study offers insights for optimizing UV-B radiation dosage to produce *L. ruthenicum* tea with increased anthocyanin content.

## MATERIALS AND METHODS

### Plant materials and UV-B irradiation treatment

Seeds used in this experiment of *L. ruthenicum* were collected from the desert area of Dulan County, Qinghai Province, China. The healthy and similar seeds were collected and germinated in a petri dish for 15 days before being transplanted into organic soil. The seedlings were then grown for 80 days in an environment with a light/dark cycle (16 h/8 h, 25/20 °C) and a humidity of 60%. Twelve plants of similar size were selected, with each group comprising four plants, resulting in a total of three groups, which were designated as three biological replicates. All data were obtained based on these three biological replicates. The samples that were not subjected to UV-B irradiation served as the control group (CK), while the samples that were irradiated with UV-B constituted the treated group.

UV-B irradiation was produced by a UV-B lamp with a main spectral line of 253.7 nm and a real-time power density of 10 $\mu$w/cm$^2$. During each light/dark cycle, the plants in the experimental group were intermittently exposed to the UV-B irradiation for 12 periods during the light cycle in each day. The irradiation duration of each period was 10 min (*i.e.,* 8:00-8:10, 9:00-9:10,..., 19:00-19:10). After being irradiated for 0.5 days (6 periods), 1 day (12 periods), 1.5 days ((12+6) periods), 2 days ((12+12) periods), and 4 days ((12+12+12+12) periods), the leaves were collected from the same batch of plants for physiological indicators and RNA-Seq analysis.

### Determination of anthocyanin content

Anthocyanin content was determined using the methods described by Neff (*Neff & Chory, 1998*; *Ai et al., 2016*). According to the laboratory optimized anthocyanin extraction method from *L. ruthenicum*. The Fresh leaves of *L. ruthenicum* (0.1 g) were ground into fine pieces and extracted with 1 mL of methanol: water (65% v/v, Methanol: >99% purity; Thermo Fisher Scientific, Waltham, MA, USA) solution (hydrochloric acid volume ratio: 1%) using ultrasonication with three replicates. The ultrasonic time was 20 min. The temperature was maintained at room temperature. The optical density of 530 nm ($A_{530}$)

and 657 nm ($A_{657}$) were measured using a microplate reader, then the anthocyanin content was calculated as ($A_{530}$−0.25*$A_{657}$)/m.

## Measuring of $H_2O_2$, malondialdehyde and antioxidant enzymes activities

The contents of malondialdehyde (MDA) were determined in the fresh leaves of *L. ruthenicum* by the thiobarbituric acid method, using MDA detection Kit (MDA-1-Y). $H_2O_2$ in leaves of *L. ruthenicum* were extracted by acetone and the contents were determined using the Kit ($H_2O_2$-1-Y). The activities of SOD, POD and catalase (CAT) were determined in the fresh leaves of *L. ruthenicum* under treatment. The three enzymes indexes were determined according to the manufacturer's protocol of assay kits SOD-1-W for SOD activity; POD-1-Y for POD activity; CAT-1-Y for CAT activity. All the kits for activities were purchased from Comin Biotechnology Co., Ltd., Suzhou, China (http://www.cominbio.com) (*Chen et al., 2022*).

## RNA-Seq analysis

Transcriptome sequencing was performed by GeneDenovo Biotechnology Co. (Guangzhou, China). Total RNA was extracted using the TIANGEN Polysaccharide Polyphenol Plant Total RNA Extraction Kit (Cat # DP441; TIANGEN Biotech, Beijing, China). mRNA was enriched using Oligo(dT) magnetic beads and further purified. The mRNA was reverse transcribed using random primers to synthesize cDNA. A library was constructed through PCR amplification, and sequencing was conducted on the Illumina NovaSeq 6000 platform.

After sequencing, raw reads were subjected to quality control to obtain clean reads. These reads were assembled using Trinity software, resulting in high-quality unigenes. The unigene sequences were aligned against the NR, SwissProt, KEGG, and COG/KOG protein databases using BLASTx (E-value<$1 \times 10^{-5}$) for functional annotation. RSEM (v1.2.19) was used to quantify the assembled unigenes. DESeq2 (v1.20.0) was used to analyze the differentially expressed genes (DEGs) between two groups. DEGs were identified using thresholds of FDR<0.05 and |log2FC| ≥2. KEGG pathway enrichment analysis was performed using KOBAS software (v2.0.12) to assess the statistical significance of DEGs in KEGG pathways.

## qPCR analysis

RNA was extracted using the Polysaccharide Polyphenol Plant Total RNA Extraction Kit (DP441; Tiangen, China). cDNA synthesis was performed using the Rapid Reverse Transcription Kit (Takara, RR092A). qPCR analysis was performed using a quantitative PCR kit (RR820A; Takara) with three replicates. The primer sequences are shown in Table S1. Actin from *L. ruthenicum* was used as the internal reference gene. The expression of genes was calculated using the $2^{-\Delta\Delta Ct}$ method.

## Interaction network analysis of structural genes and transcription factors for anthocyanin synthesis

Using the STRING protein interaction database (http://string-db.org), the Omicsmart online platform of GeneDenovo Biotechnology Co. predicted the transcription factors that

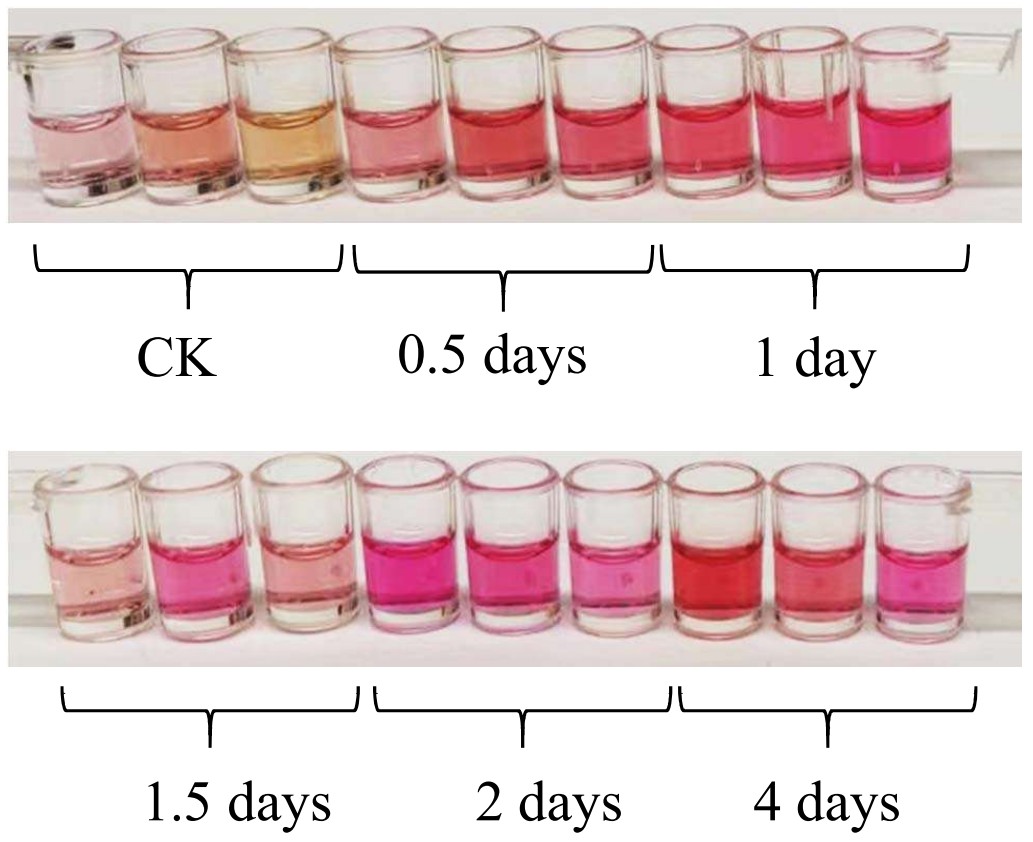

**Figure 1** **The color of the anthocyanin extraction solution with different UV-B irradiation duration.**

interact with 24 structural genes involved in anthocyanin synthesis. Out of the 12,720 genes that were differentially expressed after UV-B irradiation treatment, 424 were identified as transcription factors. An interaction network was created by using the 24 structural genes as a target gene set and the 424 transcription factors as an associated gene set. Only prediction results with a combined score >400 were retained.

## RESULTS

### UV-B induced anthocyanin accumulation and changes in antioxidant enzymes

After being exposed to UV-B irradiation for 0.5 days, 1 day, 1.5 days, 2 days, and 4 days, the color of the anthocyanin extraction solution obtained from the leaves of *L. ruthenicum* changed significantly, as shown in Fig. 1. The color of the anthocyanin extraction solution became darker after UV-B treatment compared to the control group (CK).

Figure 2A displays the anthocyanin content measured by spectrophotometry for varying durations of UV-B irradiation. The highest level of anthocyanin content was observed after 1 day of exposure to UV-B irradiation, with a relative content of 5.43 $(A_{530}-0.25^{*}A_{657})$ $g^{-1}$, which is approximately 2.8 times higher than that of the control group. However, as the duration of UV-B irradiation increased, the anthocyanin content gradually decreased.

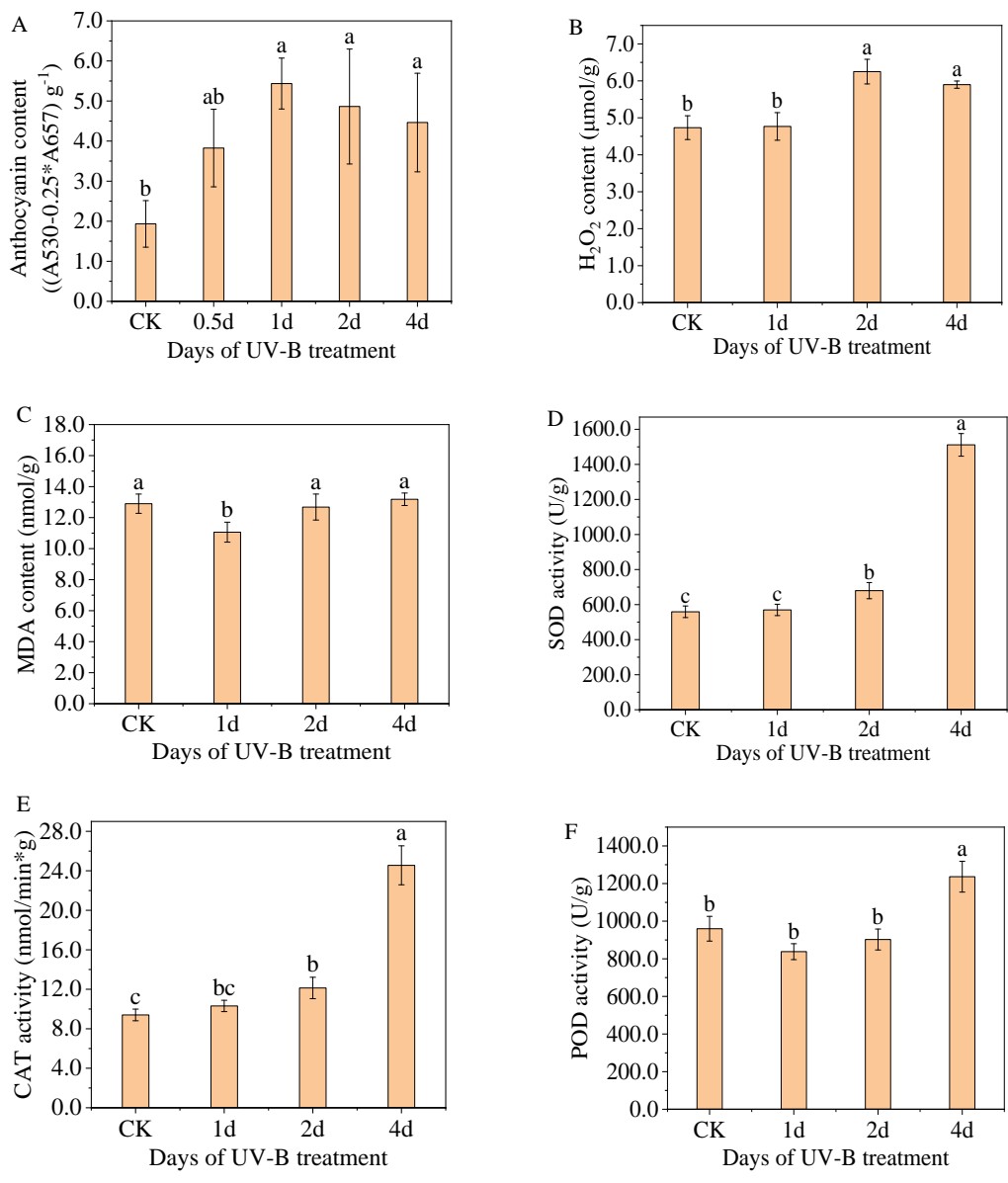

**Figure 2** (A–F) Total content of anthocyanins and antioxidant enzymes in the leaves of *L. ruthenicum Murr* after 0.5, 1, 2, 4 days of UV-B irradiation.

Because when treated for a short time, UV-B radiation can promote the resistance of plants, but after a certain threshold, the resistance will no longer increase but begin to weaken. Furthermore, Figs. 2B–2F display the changes in the content of enzymes, *i.e.,* $H_2O_2$, MDA, SOD, POD, and CAT, in the leaves of *L. ruthenicum* after UV-B irradiation treatment. The content of antioxidant enzymes (excluding MDA) increased after UV-B irradiation treatment, indicating that increasing antioxidant enzyme activity is a physiological response for plants to adapt to UV-B irradiation stress. Enhanced antioxidant enzyme activity and
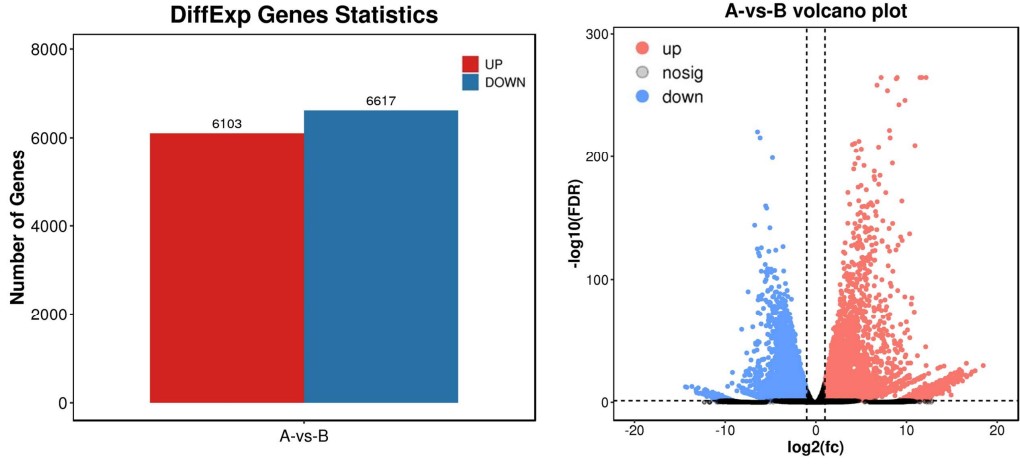

**Figure 3  Statistics of differential expression genes.**

anthocyanin synthesis represent components of the plant defense system. These responses enable plants to adapt and resist the stress caused by UV-B radiation.

## Differential expression genes after UV-B irradiation treatment

The anthocyanin content in the leaves of *L. ruthenicum* changed significantly after 1 day of UV-B irradiation treatment. Therefore, the leaves that were exposed to UV-B irradiation for 1 day were selected for transcriptome analysis. Six transcriptome libraries of *L. ruthenicum* leaves were constructed, which were then divided into a control group (CK) and a treatment group. The control group consisted of Lr-CK-1, Lr-CK-2, and Lr-CK-3, while the treatment group consisted of Lr-UV-B-1, Lr-UV-B-2, and Lr-UV-B-3.

After filtering the data and quality control, a total of 61~72 million clean reads were obtained. In our libraries, the Q20 and Q30 statistics of the clean reads were greater than 97% and 93%, respectively (Table S2). The assembly produced 63,698 Unigenes with an average length of 1,125 bp and an N50 of 1945 bp (Tables S4 and S6). Over 50% of the Unigenes had lengths ranging from 200 to 2,000 bp (Fig. S1). All unigenes were subsequently annotated using BLASTx ($E$-value $< 1 \times 10^{-5}$) searches of four public databases: NCBI nr database, Swiss-Prot protein database, KEGG database, and COG database. The annotation results are presented in Table S5. A total of 63,698 unigenes were processed and low expressed genes were removed, resulting in a total of 43,989 unigenes. The statistical power of this experimental design, calculated in *RnaSeqSampleSize* is 0.82.

To identify differential expression genes after UV-B irradiation treatment, we used DESeq2 software to generate histograms and volcano plots of differential genes, as shown in Fig. 3. The thresholds used to identify differential expression genes between each group were DFR < 0.05 and log2 FC > log2(2). As a result, we identified a total of 12,720 differential expression genes, including 6,130 genes with increased expression and 6,617 genes with decreased expression (Table S7).

To study the functions of differential expression genes, we performed KEGG functional enrichment analysis. The unigenes were divided into 140 pathways in the KEGG functional

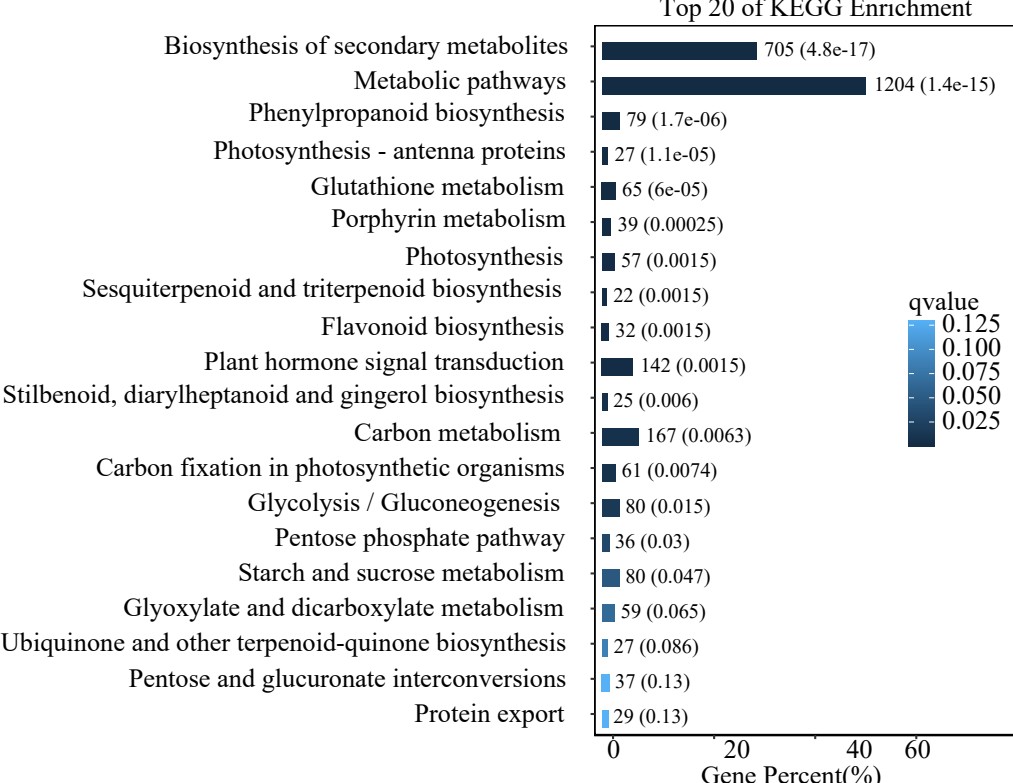

**Figure 4 Top 20 enriched KEGG pathways of DEGs.**

classification, with the top 20 pathways shown in Fig. 4. The metabolic pathway were the most significantly enriched with 1,204 unigenes, followed by the biosynthesis of secondary metabolites pathway with 705 unigenes. Other enriched pathways include phenylpropanoid metabolism, flavonoid metabolism, and hormone signal transduction. These pathways play crucial roles in the leaves of *L. ruthenicum* response to UV-B stress by regulating the synthesis of protective compounds and mediating stress signaling.

## Analysis of genes related to UV-B signal transduction

The differential expression genes in the UV-B signaling pathway were identified. The expression of seven genes was observed to change: one COP1 gene (Unigene0008780), one HY5 gene (Unigene0072293), and five UVR8 genes (Unigene0015353, Unigene0003921, Unigene0067978, Unigene0020095 and Unigene0078284). Specifically, following exposure to UV-B irradiation, the majority of UVR8 genes exhibited a decrease (Fig. 5A), while the Unigene0067978 demonstrated a significant increase. In contrast, the COP1 genes exhibited a decrease, while the expression of HY5 increased.

The results of the qPCR analysis indicated a significant increase in the expression of UVR8 (Unigene0067978) following one day of UV-B treatment, which then decreased (Fig. 5B). The expression of COP1 initially decreased and then increased significantly with

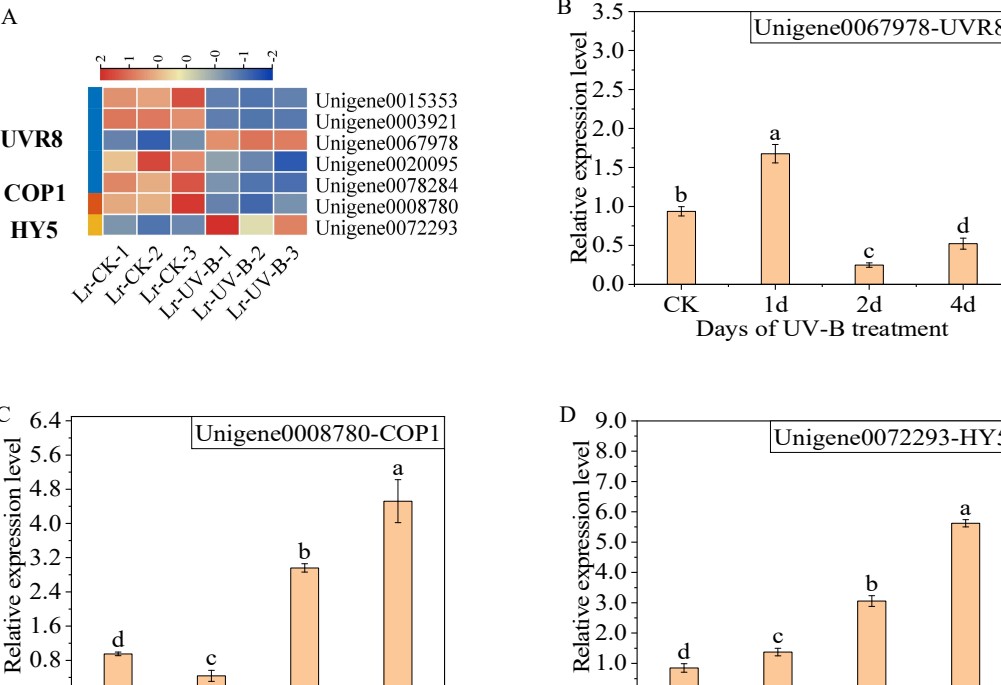

**Figure 5 Expression heat map in the UV-B signaling pathway after UV-B irradiation and RT-qPCR verification of the differential expression genes.** (A) Expression heat map. (B–D) The RT-qPCR results of UVR8, COP1 and HY5.

the duration of treatment (Fig. 5C). The expression of HY5 increased continually with the treatment duration (Fig. 5D).

## Analysis of genes related to anthocyanin synthesis

We analyzed the unigenes involved in the synthesis pathways of flavonoids (Fig. 6A) and anthocyanins under UV-B irradiation. A total of 24 enzyme-encoding genes were identified and their expression was found to be up-regulated, as shown in Fig. 6B. Among the identified genes, the two genes with the most significant increase in expression were F3'H (Unigene0009145, FPKM is 891 times that of CK) and C4H (Unigene0046607, FPKM is 18 times that of CK).

To ensure the reliability of the RNA-seq data, we selected CHS (Unigene0010556) and UFGT (Unigene0001601), which are enriched in the anthocyanin biosynthesis pathway (ko00942), for qPCR analysis. The results showed a significant increase in the expression of CHS (Unigene0010556) following one day of UV-B treatment (Fig. 6C). However, the expression decreased as the duration of UV-B irradiation treatment increased. The expression of UFGT (Unigene0001601) initially increased and then decreased with UV-B treatment time, as shown in Fig. 6D. The results of the qPCR analysis were consistent with the transcriptome analysis results.

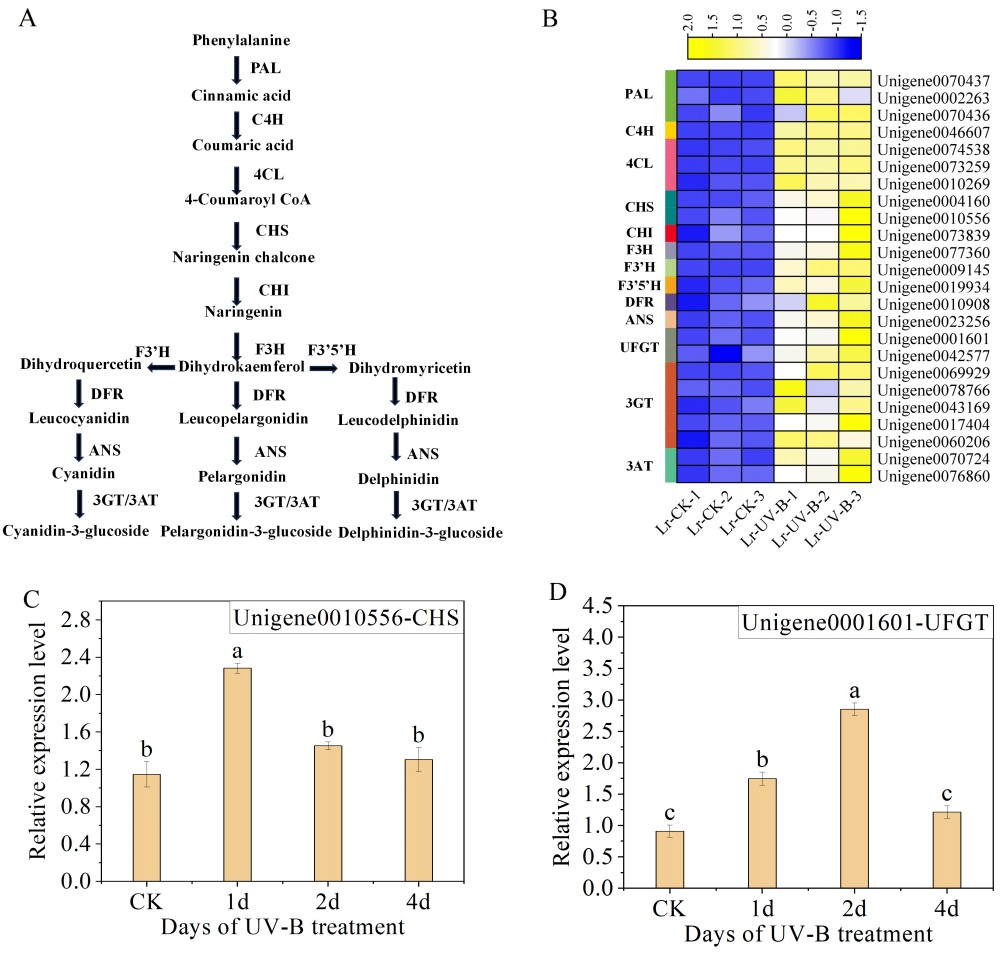

**Figure 6  Expression heat map and RT-qPCR verification of the DEGs in the anthocyanin biosynthesis.** (A) Simplified scheme. (B) Expression heat map. (C and D) The RT-qPCR results of CHS and UFGT.

## UV-B signal transduction and anthocyanins synthesis transcription factors

A total of 424 transcription factors, belonging to 36 gene families, were identified among the differential expression genes. The largest family group was MYB transcription factors (15%), followed by ERF (14%), bHLH (10%), and WRKY (10%). The changes in expression of MYB, bHLH, and WRKY transcription factors were examined. The expression of most transcription factors was found to decrease in response to UV-B irradiation, as shown in Fig. 7. However, some transcription factors showed a significant increase in expression, such as WRKY1 (Unigene0066868, FPKM value 100 times higher than that of CK), WRKY2 (Unigene0013893, FPKM value 63 times higher than that of CK), WRKY3 (Unigene0039337, FPKM value 47 times higher than that of CK), and WRKY4 (Unigene0078996, FPKM value 14 times higher than that of CK).

The qPCR analysis was conducted with the following transcription factors: MYB (Unigene0070868), bHLH (Unigene0025729), and WRKY (Unigene0066868, Unigene0013893,

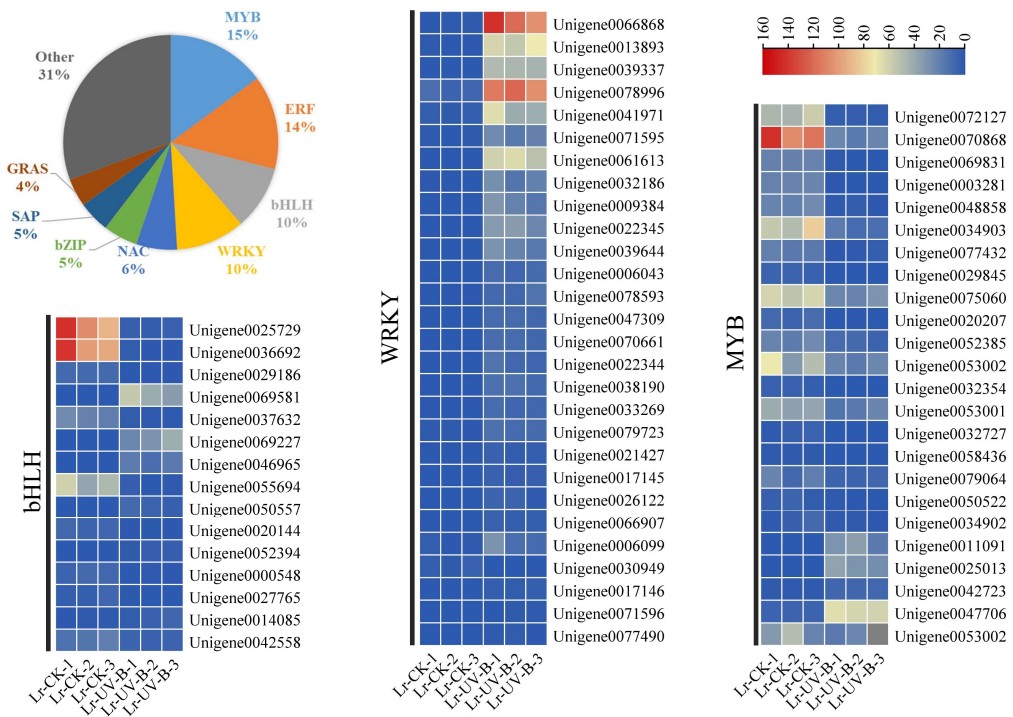

**Figure 7 Classification and proportion of differential expression transcription factors and expression heat map.**

Unigene0039337, Unigene0078996). The results are presented in Fig. 8, which demonstrates that the change in the expression of the six transcription factors, with the exception of MYB, is consistent with the transcriptome analysis results. Furthermore, the expression of WRKY transcription factors significantly increased following exposure to UV-B irradiation. This indicates that WRKY family genes may play a crucial role in responding to UV-B irradiation.

## Network analysis of interactions between structural genes and transcription factors in anthocyanin biosynthesis

To study the regulatory role of transcription factors in anthocyanin biosynthesis, we analyzed the interactions between 24 structural genes involved in anthocyanin synthesis and transcription factors. A network graph was constructed based on protein-protein interactions with a combined score above 400. As shown in Fig. 9A, a total of 12 structural genes involved in anthocyanin synthesis formed 40 pairs of interactions with 11 transcription factors, all of which exceeded the score threshold. Notably, MYB1 (Unigene004770600) showed the highest connectivity, interacting with all 11 structural genes. Following MYB1, bHLH (Unigene0014085) and MYB308 (Unigene0053002) interacted with 9 and 5 structural genes, respectively. Among the structural genes, the CHS genes (Unigene0004160, Unigene0010556) showed the highest connectivity, interacting with 6 transcription factors, followed by ANT17 (Unigene0004160). As shown in Fig. 9B,

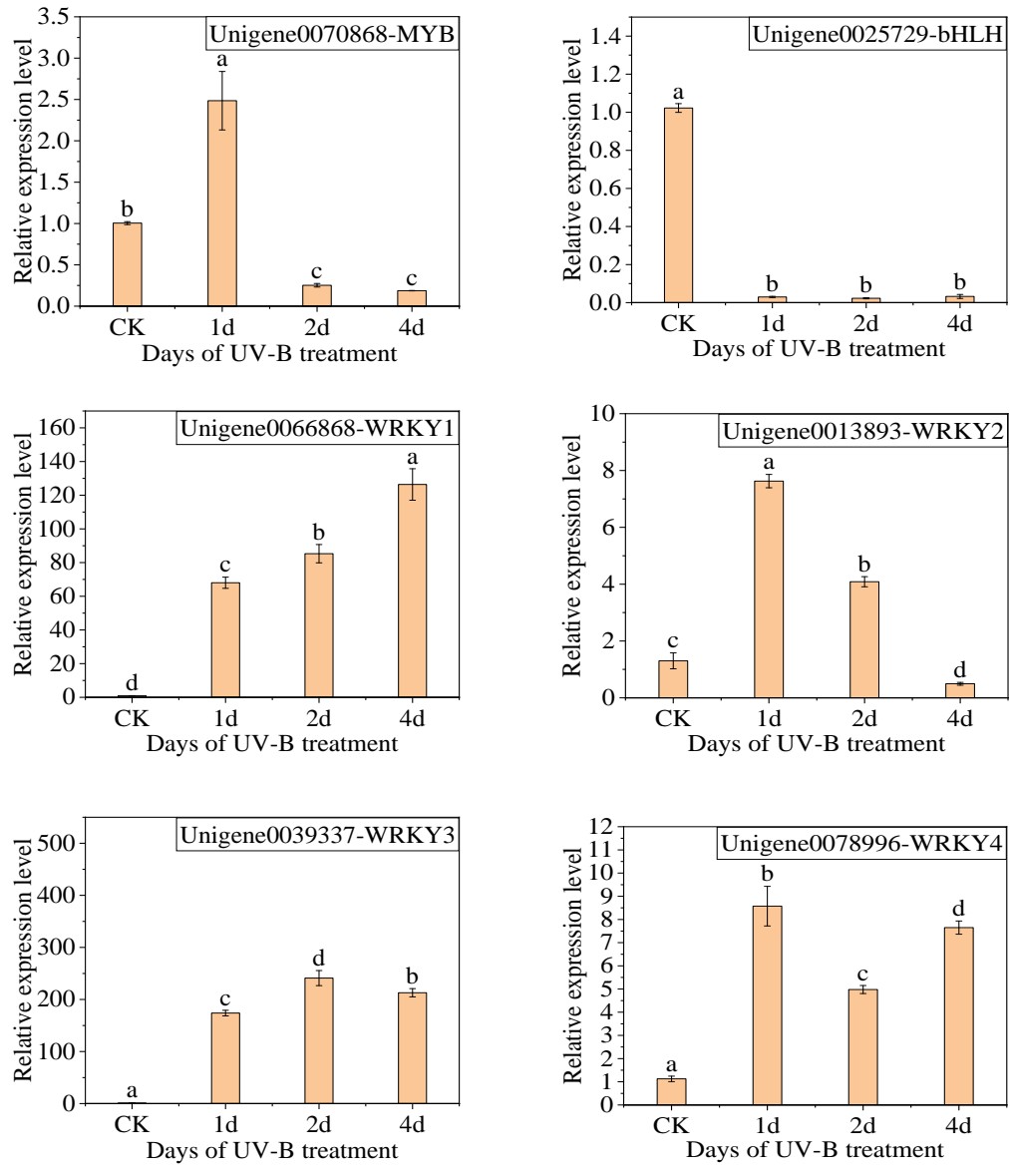

**Figure 8  The results of RT-qPCR for the transcription factors.**

most of the interacting transcription factors showed up-regulated expression, with the exception of JAF13, MYB4, and TAF1.

## DISCUSSION

Our findings indicate that the leaves of *L. ruthenicum* are abundant in anthocyanins, with a significant increase in content observed following exposure to UV-B irradiation. This suggests that anthocyanin content may be a direct response of the leaves of *L. ruthenicum* to UV-B irradiation (*Del et al., 2015*). Studies on many other plants have also shown an increase in anthocyanin content following UV-B irradiation. These include *Populus alba*

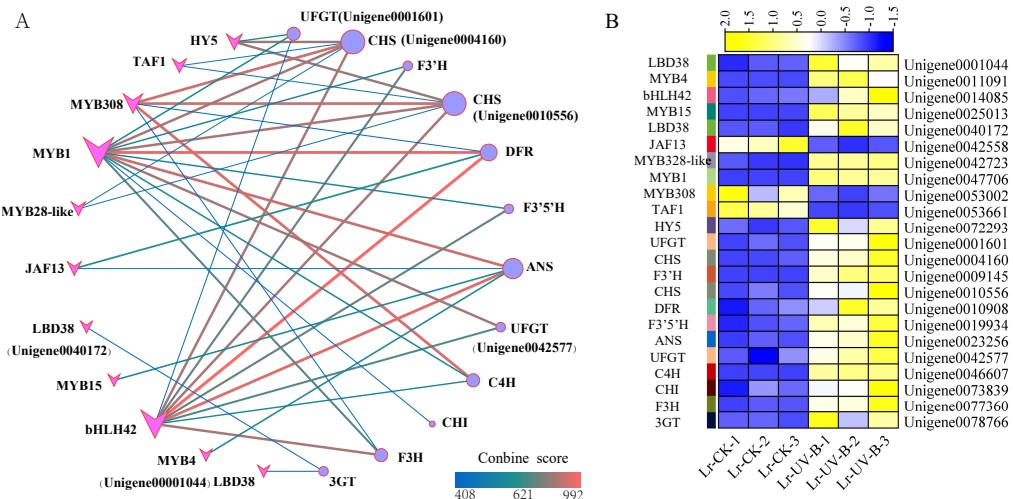

**Figure 9** **Protein-protein interaction networks and heat map of their expressions.** (A) Interaction network between structural genes of anthocyanin synthesis and transcription factors. (B) Expression heat map of genes.

and *Populus russkii* (*Ma et al., 2016*), *Artemisia annua L.* (*Pandey & Pandey-Rai, 2014*), and *Indigofera tinctoria* (*Ravindran et al., 2010*).

The molecular mechanism of anthocyanin biosynthesis in the leaves of *L. ruthenicum* following UV-B irradiation was investigated by transcriptome testing. The analysis revealed that the 24 identified structural genes in the anthocyanin synthesis pathway were up-regulated, resulting in an increase in anthocyanin content. Furthermore, the study analyzed the genes expressed on the UV-B signaling pathway. The results demonstrated that following UV-B irradiation, UVR8 (Unigene0067978), COP1, and HY5 were up-regulated. This result is consistent with the expression patterns of UVR8, COP1 and HY5 in *Arabidopsis thaliana* and *Ginkgo biloba* leaves (*Favory et al., 2009*; *Yang et al., 2020*) .

The expression of WRKY2 increased significantly and reached its highest level after 1 day of UV-B irradiation treatment. As previously stated, the anthocyanin content in the leaves of *L. ruthenicum* also reached its highest level after 1 day of UV-B irradiation treatment. Therefore, the trend of anthocyanin content changing with UV-B irradiation treatment duration was consistent with the changing trend of WRKY2 expression. This suggests that the WRKY2 transcription factor positively regulates anthocyanin synthesis in *L. ruthenicum* leaves in response to UV-B irradiation. Similar results have also been obtained in *apples* (*Hu et al., 2020*). They likely function by binding to specific cis-acting elements in the promoters of target genes related to anthocyanin biosynthesis, thereby activating their transcription under UV-B stress conditions (*Liu et al., 2019*). Specifically, WRKY transcription factors may modulate the expression of key enzymes in the anthocyanin biosynthetic pathway, such as phenylalanine ammonia-lyase (PAL) and chalcone synthase (CHS), enhancing anthocyanin accumulation as a protective response against UV-B-induced oxidative damage (*Su et al., 2022*). Moreover, WRKY factors can interact with other transcriptional

regulators and signaling components, potentially forming a regulatory network that coordinates plant responses to UV-B stress (*Lloyd et al., 2017*).

The prediction of transcription factors that interact with structural genes revealed that MYB1 had the highest connectivity and interacted with 11 structural genes. MYB1 was identified as an AN2-like protein (*Li et al., 2020*), a transcription factor that plays a crucial role in regulating anthocyanin synthesis in *L. ruthenicum*. Therefore, MYB1 may be a key transcription factor in regulating anthocyanin synthesis in *L. ruthenicum* leaves following UV-B radiation. Additionally, we found that HY5 interacts with two CHS genes. Previous research has demonstrated that under UV-B irradiation, HY5 can bind to the promoter of CHS and activate its transcription (*Ang et al., 1998*). Furthermore, studies have shown that HY5 protein can bind to the CHS promoter sequence both *in vitro* and *in vivo* (*Lee et al., 2007*). Therefore, HY5 and CHS are also considered key genes for anthocyanin synthesis.

## CONCLUSIONS

A twofold increase in anthocyanin content was observed in the leaves of *L. ruthenicum* following a one-day UV-B irradiation treatment. The expression of 24 structural genes identified in the anthocyanin synthesis pathway was up-regulated. The transcription factors MYB1 and bHLH were suggested to have potential roles in regulating anthocyanin synthesis. Additionally, UVR8, HY5 and COP1 were found to be highly involved in the transduction of UV-B signals.

The findings presented here provide a foundation for future molecular biology studies on the promotion of anthocyanin accumulation in *L. ruthenicum* under UV-B irradiation. Future studies may explore the specific molecular mechanisms by which these genes regulate anthocyanin synthesis. In conclusion, this study enhances our understanding of how specific genes and pathways are regulated in response to UV-B exposure, laying the foundation for potential applications in improving plant fitness and nutritional quality.

## ACKNOWLEDGEMENTS

The authors would like to thank Dr. Zhanwen for his help and anonymous reviewers for their helpful comments.

### Funding

This work was supported by the Youth Fund of Qinghai Provincial Department of Science and Technology (No. 2024-ZJ-988). The funders had no role in study design, data collection and analysis, decision to publish, or preparation of the manuscript.

### Grant Disclosures

The following grant information was disclosed by the authors:
The Youth Fund of Qinghai Provincial Department of Science and Technology: 2024-ZJ-988.

### Competing Interests

The authors declare there are no competing interests.

### Author Contributions

- Shengrong Chen conceived and designed the experiments, performed the experiments, analyzed the data, prepared figures and/or tables, authored or reviewed drafts of the article, and approved the final draft.
- Yunzhang Xu conceived and designed the experiments, prepared figures and/or tables, and approved the final draft.
- Weimin Zhao analyzed the data, prepared figures and/or tables, and approved the final draft.
- Guomin Shi conceived and designed the experiments, prepared figures and/or tables, and approved the final draft.
- Shuai Wang performed the experiments, prepared figures and/or tables, and approved the final draft.
- Tao He conceived and designed the experiments, authored or reviewed drafts of the article, and approved the final draft.

### Data Availability

The raw data are available in the Supplementary Files.

### Supplemental Information

Supplemental information for this article can be found online at http://dx.doi.org/10.7717/peerj.18199#supplemental-information.

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
