# Peer review of "UV-B irradiation promotes anthocyanin biosynthesis in the leaves of Lycium ruthenicum Murray"

_PeerJ, doi:10.7717/peerj.18199_

## Round 0.1 · original submission · Minor Revisions

Dear Dr. He,

Please read carefully all the three detailed reviews to improve your manuscript. Particularly important the methods should be revisited .e.g. Reviewer #1 states "Chemicals used in this research should be given with the purity, company, and country. Extraction parameters such as solid:solvent ratio, temperature, and time should be stated in more detail. The method of determining total monomeric anthocyanin content (TMAC) should be rechecked. It commonly uses the pH differential method". I also observed that the RNa-seq analysis part is quite short and should be better explained. Thus I ask you to provide a point-by-point answer to all points raised by the three reviewers.

·

Basic reporting

The figures and tables are provided in this paper but it would be easier to follow when they are close to the texts. Some related studies need to be added to support the findings.

Experimental design

Chemicals used in this research should be given with the purity, company, and country. Extraction parameters such as solid:solvent ratio, temperature, and time should be stated in more detail. The method of determining total monomeric anthocyanin content (TMAC) should be rechecked. It commonly uses the pH differential method.

Validity of the findings

Figure 1 would be more helpful when it is supplemented with their UV-Vis Spectra showing the maximum wavelength and absorbance. The conclusion needs to be revised, stating the main finding to answer the research question.

Additional comments

Some typos are found in the draft, and they have been highlighted in the text. The explanation for each finding related to anthocyanins' accumulation in leaves affected by UV-B should be discussed in more detail.

Reviewer 2 ·

Basic reporting

attached

Experimental design

attached

Validity of the findings

attached

Annotated reviews are not available for download in order to protect the identity of reviewers who chose to remain anonymous.

·

Basic reporting

No

Experimental design

No

Validity of the findings

No

Additional comments

Review report

Overall, this research article represents an interesting “UV-B irradiation promotes anthocyanin biosynthesis in the leaves of Lycium ruthenicum Murray”. Contains a relatively short introduction. I suggest authors increase its length. Overall, the manuscript is interesting and well-structured. Moreover, the Materials and methods & results are clearly described making the manuscript understandable for readers. In order to improve the present study, Minor revision has to be done and some essential modifications have to be fixed before it proceeds, and decisive action can be taken. In addition, the study needs some editing on some minor grammatical issues. And follow the paper guideline. All the comments, remarks and suggestions are given below.

Abstract.
1: The findings are well-summarized, but the implications of the study could be expanded. How do these findings contribute to the broader field of molecular biology or agricultural science?
2: Ensure consistent use of scientific names throughout the paper. For example, "Lycium ruthenicum Murray" is used in some parts, while "L. ruthenicum Murr." is used in others. Standardize this for clarity
3: The role of the protein interaction network is mentioned briefly. Could you clarify how this network analysis contributes to understanding UV-B signal transduction?

Introduction:
4: The introduction mentions a lack of systematic research on anthocyanin synthesis in L. ruthenicum Murr. under UV-B irradiation. This gap could be emphasized more strongly to underscore the novelty and necessity of the study. Can you elaborate more on why this gap is critical?
5: Have other studies explored UV-B effects on similar plants? How does your approach differ from or build upon these studies?
6: Author should describe the (COP1 and HY5) in introduction.





Results:
Lines 141-144: You noted that anthocyanin content peaked at 1 day and then decreased with longer UV-B exposure. Do you have any hypotheses or explanations for this pattern? How might prolonged UV-B exposure affect anthocyanin degradation or synthesis?

Lines 174-175: The mention of enriched pathways includes some repetition ("Other enriched pathways include phenylpropanoid metabolism, flavonoid metabolism, and hormone signal transduction."). Consider revising to avoid redundancy.

Lines 178-183: The expression of UVR8 and COP1 genes showed different patterns. What might account for the initial decrease and subsequent increase in COP1 expression? Are there known feedback mechanisms or regulatory processes that could explain this?

Lines 203-219: The results highlight the significant increase in WRKY transcription factors. Could you discuss the potential functional roles of these WRKY transcription factors in UV-B response and anthocyanin synthesis?

---

## Round 0.2 · Minor Revisions

Dear Dr. Chen,

Please consider the observations below, from one of the Section Editors, and perform the changes, and provide a point-by-point reply.

"RNAseq details need to be clarified and data needs to be deposited. Details:

1) The RNAseq methods are unclear. Specifically, the Methods section says that RSEM was used to determine expression levels but does not say how differentially expressed genes were determined. The Results sections states that DEseq was used. Was DEseq used or RSEM? Please clarify in methods.

2) Line 200 mentions "DFR". Is that a typo for "FDR"?

3) All RNAseq reads need to be deposited to a sequencing data repository.

4) The methods state that "reads were assembled using Trinity software, resulting in high-quality unigenes". The authors need to provide assembly data...how many transcripts? average length? BUSCO statistics? etc.

5) assembled transcripts (with annotation) should be deposited into a data repository or included as supplementary material.

6) A table of results from DEseq (or whatever was used) giving the FDR and LFC for every gene should be included as supplementary material."

·

Basic reporting

No comment

Experimental design

No comment

Validity of the findings

No comment

---

## Round 0.3 · accepted · Accept

Dear Dr. He,

Thanks for addressing all the comments.

Congratulations